# Video Caching at Data-drifting Network Edge: A KD-based Cross-domain Collaborative Solution

## Abstract

The surge in video streaming has caused network congestion and quality decline, posing a significant challenge for efficient content delivery. Edge caching, using mobile edge caching servers like edge base stations (EBS), has emerged as a cost-effective solution. Collaborative edge caching, addressing space limitations by fostering cooperation and content sharing among servers, improves caching hit rates (CHR). However, little attention has been paid to the impact of request characteristics on different servers. To tackle this issue, we conducted a study using data collected from Kuaishou company over a period of four weeks, comprising 350 million video requests. Our research findings indicate that request-sparse EBSs significantly impede the overall CHR improvement. Knowledge distillation (KD), a technique that transfers knowledge from strong models to weak models, is expected to solve this bottleneck problem. However, traditional KD methods often rely on the assumption of independent and identically distributed data, which may not hold true in real-world scenarios where data drift occurs. We identify two major types of data drift in caching data: temporal drift and spatial drift. To overcome these challenges, we propose an adaptive KD-based cross-domain collaborative edge caching (KDCdCEC) framework, incorporating three tailored components: i) a slot-wise DRL-based KD-enhanced caching agent capable of adapting to EBSs with varying storage sizes, ii) a deep deterministic policy gradient-based algorithm that adaptively configures the reference weights of EBS on their KD partners, and iii) a content-aware request routing mechanism for partner adjustment. Experimental results show that KDCdCEC outperforms state-of-the-art approaches in average CHR, average latency, and traffic cost.

## 1 Introduction

Video streaming, including live streams and clips, is projected to comprise over 65% of global internet traffic by the end of 2023, as reported by Sandvine Sandvine (2023). This rapid growth has caused network congestion and potential quality decline, posing a challenge for content providers. Edge caching, often through a content delivery network (CDN), aims to address this issue by enhancing delivery efficiency. However, CDNs struggle to adapt to bursty content during peak periods.

With the development of mobile communication technologies, such as 5G, mobile edge caching has introduced a novel solution for video content delivery by bringing the content closer to users with lower cost Hu et al. (2015). However, the storage space of mobile edge caching servers (*i.e.,* edge base stations, abbreviated as EBSs) is limited, impacting the increase of CHR. Collaborative edge caching has recently emerged as a promising solution to address this limitation Poulimeneas et al. (2016); Nomikos et al. (2021); Tran & Pompili (2018). Collaborative edge caching, through EBS cooperation and content sharing, boosts CHR, leading to decreased average latency and traffic costs.

Despite notable progress in edge caching technologies, scant attention has been given to the impact of request characteristics on various servers. To bridge this gap, we conducted a comprehensive study using data collected from Kuaishou company[1] over a four-week period, encompassing 350

---

[1]Kuaishou stands as one of the premier short video platform of China, boasting a daily active user base of 355.7 million by the end of 2022Thomala (2023).

million video requests, revealing substantial variations in the number of requests handled by different EBSs in real-world scenarios (shown in Fig. 1(a))[2]. Specifically, at least 78% of EBS received fewer user requests compared to the midrange. Additionally, we noted a strong positive correlation between model CHR and the number of requests handled by various servers. (coefficient of determination $R^2 = 0.8671$). Furthermore, the average CHR of request-sparse EBSs was 45% lower than that of request-dense EBSs (request volume higher than midrange). Therefore, *request-sparse EBSs significantly bottleneck the overall CHR improvement in the edge caching problem.*

Recently, knowledge distillation (KD) Gou et al. (2021) has proven effective in transferring valuable insights from a strong model to a weaker one. In the context of edge caching, we assert that models on request-dense EBSs harbor richer knowledge for decision-making, guiding cache replacement for other EBSs. To tackle this, we propose a KD-based collaborative solution for video caching.

However, traditional KD methods often rely on the independent and identically distributed assumption, which may not hold in real-world scenarios with data drift. Through our measurements, we identified two major types of data drift in caching data: **temporal drift** and **spatial drift**. Temporal drift refers to the rapid changes in the popularity of requested videos over time. To quantify this phenomenon, we defined three metrics: request frequency (number of requests per minute for a single video), request proportion (ratio of request frequency to total request volume in the corresponding period), and request similarity (proportion of the same requested video to all requested videos in two different minutes). We randomly select various time periods and observe that, on average, the request frequency of all requested videos changed by 34% after 2 hours, while the average request similarity after 24 hours was only 59%. Different video types exhibit distinct frequency change trends. Spatial drift, on the other hand, relates to the unevenly distributed requested content of different geographical domains[3]. We use the coefficient of variation (CV) as a standardized measure of dispersion for request distribution. After selecting various time periods randomly, we observe that the request proportion CV for 56% of videos in each domain exceeded 1 on average[4]. Additionally, the average CV of the request frequency change ratio after 24 hours in each domain for all requested videos was 1.18. More measurement results are shown in Fig. 1.

Given the observations, we aim to address the following challenge: *Facing the complicated imbalance requests of EBS, how to develop an effective collaborative framework against temporal and spatial drift simultaneously?* To achieve this goal, we present an **adaptive KD-based cross-domain collaborative edge caching framework, called KDCdCEC**, involving three components.

Firstly, i): Motivated by the great success of deep reinforcement learning (DRL) on the temporal drift problem, we formulate edge caching as a Markov decision process (MDP) and integrate it with KD to facilitate collaborative edge caching. To reduce the computational complexity for EBSs with large storage space, we design a slot-wise reinforcement learning agent that can be directly applied to EBSs with different storage sizes. ii): To handle spatial request drift, we propose a deep deterministic policy gradient (DDPG)-based algorithm that adaptively configures the reference weights of servers on their partners. iii): We recognize that the content characteristics of requests can differ significantly between EBSs and their partners. As such, dynamically adjusting weights may impact the performance of intermediate processes. To mitigate this issue, we introduce a content-aware request routing mechanism that can directly discard unsuitable partners, thus enhancing the previous component. We sampled about half of the Kuaishou dataset for experimental evaluation. In comparison to the best-performing collaborative baseline, KDCdCEC can improve CHR by 5.26% with less cost (0.94% latency reduction and 5.16% traffic cost reduction).

## 2 PRELIMINARY

### 2.1 ANALYSIS OF VIDEO REQUEST MEASUREMENT

For in-depth insights into request heterogeneity and dynamics, we analyzed 350 million real-world traces, dividing the area into a $7 \times 8$ grid for examination.

---

[2] Measurements are based on the result of non-collaborative approach in Table 3.

[3] Each EBS serves a distinct region referred to as a domain, with no overlap between them.

[4] A CV greater than 1 indicates a high degree of variability relative to the mean of the dataset and the dataset is not normally distributedAbdi (2010).

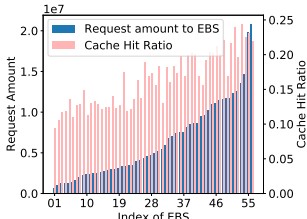 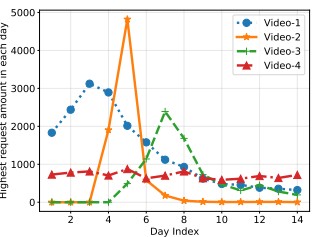 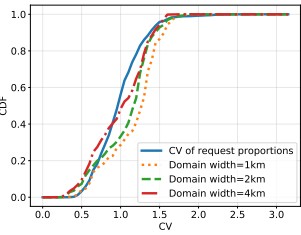

(a) Volume Drift: Request amount and CHR of EBSs

(b) Temporal Drift: Popularity trend of 4 typical videos in two weeks

(c) Spatial Drift: CV of the request frequency change ratio of all content after 24 hours in each domain

Figure 1: Real-world video request measurements and analysis.

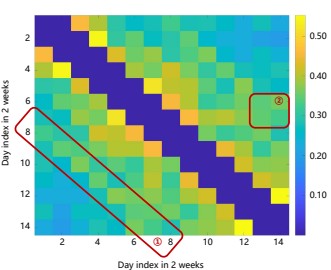

Figure 2: Daily request similarity in 14 consecutive days. The first framed pattern indicates user requests have higher similarity on the same day of the week. The other framed pattern reveals that user requests show higher similarity on weekends.

Fig. 1(a) shows the positive correlation between request amount and EBS performance as mentioned in introduction. In addition, we observe the spatio-temporal characteristics of user traces. We randomly sampled 512 videos and counted their highest daily requests for 14 consecutive days to construct daily highest request vector as feature. We adopt cosine similarity to calculate the distance between videos and classify them into 4 clusters according to the average distance scree plot. Subsequently, we select 4 representatives from the top-100 most requested videos, who are the closest to the center of mass of each cluster respectively. The daily highest request of these representatives are depicted in Fig. 1(b), which reveals that different types of content exhibit varying popularity trends. Fig. 1(c) shows the request proportion CV for 56% of videos in each domain exceeded 1 on average, indicating the proportion of more than half content varies greatly among different EBSs at the same moment. Apart from that, we change the domain width from 1km to 4km and calculate the request frequency change ratio of all content. The result attests the average CV of the request frequency change ratio after 24 hours in each domain for all requested videos is 1.18 when the domain width is set to 2km. This insight reveals the trend of the same content in different domains is generally inconsistent. Overall, our measurements highlight the rapid and dynamic nature of content popularity, showcasing diverse preferences and trends among different EBSs.

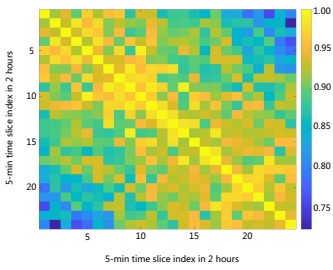

Figure 3: Request similarity in each 5-min period in 2 hours.

According to our measurement of daily content similarity in 14 consecutive days, we found two distinct features, which are framed with rounded rectangular box, as marked in Fig. 2. Furthermore, we also measured the similarity of the content of EBS requests over time. We divide requests within two hours with five-minute intervals and calculate the request similarity in each slot. As illustrated in Fig. 3, after one hour, the similarities drop to 0.85, two hours later, the similarity decay below 0.75. This result indicates content similarities of each EBS are rapidly decaying. These insights derived from our data measurements provide significant motivation for the design of our algorithm, which will be elaborated on in Section III.

## 2.2 KD-BASED EDGE CACHE FRAMEWORK

We consider a KD-based edge caching scenario for mobile video streaming, as illustrated in Fig. 4. We assume that the CDN server have cached all videos and connect to EBSs via backbone networkMa et al. (2017a). EBSs are distributed in a citywide area equipped with edge servers, which provide minor storage capacity for video caching and DRL-based model for cache decision-making. Each EBS serves the local video requests within its coverage area with two statuses: cache hit and miss, with the following steps. When a request arrives, if cache hit, EBS returns the cached content immediately (The short-dashed arrows labeled prestore and 1), or it will fetch the content from CDN

server (The long-dashed arrows labeled 1, 2, and 3). In our context, time is divided into continuous caching periods, where each EBS conducts cache replacement only at the end of each period. EBSs, communicating through a 5G network Wang et al. (2014), can collaborate and periodically acquire knowledge from partners and strengthen their own cache replacement strategies through KD (The double sided arrow between EBSs). We set the CHR as the corn optimization target, which is a measurement of how many content requests an EBS can fill successfully compared to how many requests it receives.

## 3 FRAMEWORK AND SYSTEM MODEL

### 3.1 SLOT-WISE DRL-BASED KD-ENHANCED COLLABORATIVE CACHING

Following the past works Zhong et al. (2020a); Wang et al. (2020); Ye et al. (2021); Kirilin et al. (2019), we formulate the cache replacement as MDP, and use a DRL agent to solve it.

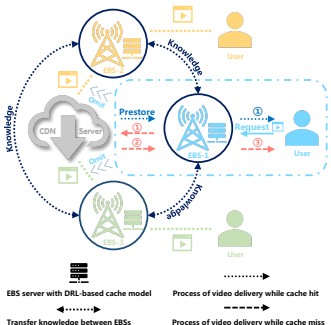

**State** States, serving as input environmental information (i.e., cache condition), are represented here by request statistics of cached videos. Specifically, we introduce an $F$-dim video feature $s_t^i = (s_t^i[w_1], \cdots, s_t^i[w_F])$, where $s_t^i[w_j]$ denotes the number of times the video in cache slot $i$ was requested in the past period $w_j$, e.g., 10s, 20s, and 40s. We set the cache capacity as $C$ [5]. Additionally, the agent considers $C$ candidate videos that possess top-$C$ feature modulus among all ever requested but un-cached videos. The state of both the cached and candidate videos constructs $s_t$, which has a dimension of $2C \times F$.

Figure 4: The conceptual flow of edge caching.

**Action** When a cache miss occurs, the EBS must determine which cache slot should be replaced with a candidate video. All possible replacement actions can be represented by a $C \times C$ one-hot binary matrix $a_t$, where the element $a_{ij} = 1$ indicates the action of replacing the video in the $i$-th cache slot with the $j$-th candidate video.

**Reward** Our target is to optimize cache space utilization by increasing cache hit times. The reward function is determined based on the iterative cache hit times after each replacement action $a_t$:

$$r_t = r_t^{i_{cache}j_{cand}}, \tag{1}$$

where $r_t^{i_{cache}j_{cand}}$ is the cumulative hit times of the $i$-th cache slot, which is filled by the $j$-th candidate video, between the action time $t$ and the next adjacent cache miss.

**Online Training** In a single-step action process, the DRL agent estimates the expected rewards of the cached and candidate content. The agent will act and continue to respond user requests until the next cache miss occurs, and then obtain the hit times and feed it back as a return. In addition, the agent will observe the current state $s_{t+1}$, then construct a transition tuple $(s_t, a_t, r_t, s_{t+1})$, and store it in the replay buffer $D$. Every once in a while, the agent samples a batch of transition tuples $\mathcal{B} = \{(s_t, a_t, r_t, s_{t+1})\}$ from the replay buffer $D$ for model training. After sampling, we apply deep Q-learning (DQN) Mnih et al. (2015), a model-free method, to design the DRL agent and construct deep neural network $Q_\theta(\cdot)$ to score contents based on their states. The loss is subsequently calculated using the sampled tuples:

$$L(\mathcal{B}|\theta) = \frac{1}{\|\mathcal{B}\|} \sum_{\{(s_t, a_t, r_t, s_{t+1})\} \in \mathcal{B}} \left[Q_\theta(s_t, a_t) - y^i\right]^2, \tag{2}$$

where $y^i = r_t + \gamma Q_{\bar{\theta}}(s_{t+1}, a)$ represents the expected discount reward, with $\gamma$ being the discount factor. $\bar{\theta}$ represent the parameter of the target network, which is initialized with the same parameter as $Q_\theta$ and updated by soft moving average. The action $a$ is generated as follows:

$$\pi_\theta(a_{t+1}|s_{t+1}) = \arg\max_{a_{t+1}} (Q_\theta(s_{t+1}, a_{t+1})). \tag{3}$$

---

[5]Following Zhong et al. (2020a); Cui et al. (2023) , we assume all videos have the same size, and the cache capacity $C$ can be represented as the number of videos that can be stored at a EBS.

Afterward, the network parameters can be updated using the following loss:

$$\theta \leftarrow \theta - \alpha \nabla_\theta L\left(\mathcal{B}|\theta\right). \tag{4}$$

The target network is updated through soft moving average:

$$\bar{\theta} \leftarrow \lambda\theta + (1 - \lambda)\bar{\theta}. \tag{5}$$

**Slot-wise transformation** We assume that the time taken to retrieve video contents from the cache is significantly shorter than their delivery time and can be neglected. Thus, different orders of the same contents in the cache can be considered as the same storage state. We simplify the cache replacement problem at the granularity of the cache slot, separating the cached content from its storage location. This simplification effectively reduces the complexity of the network input. Furthermore, it enables training an agent capable of handling the cache replacement problem for EBSs with varying cache capacities, which is particularly crucial for facilitating collaboration between EBSs.

Based on the insight from Fig. 1(a), we adopt online KD to let EBSs share knowledge for better caching decisions. Each EBS distills knowledge from the model to share with other EBSs while also enhancing its own model by acquiring knowledge from other EBSs. Specifically, we begin by extracting the sample of knowledge distillation, which requires $EBS_k$ to randomly sample a batch of transition tuple $\mathcal{S}_k = \{(s_t, a_t, r_t, s_{t+1})\}$ from the replay buffer $D$. Since each action of cache replacement involves only one cache slot, the state of other cache slots remains unchanged. Therefore, during the distillation process, the agent only needs to consider the $i$-th component of the transition if the content in the $i$-th slot is replaced, $\mathcal{S}_k$ can be simplified as $\bar{\mathcal{S}}_k = \{(s_t^i, r_t^i, s_{t+1}^i)\}$.

Next, $EBS_k$ needs to extract the sample from the tuple set by inputting $s_t^i$ into the popularity network $Q_{\theta_k}(\cdot)$ and obtains knowledge labels $Q_{\theta_k}(s_t^i)$. We can formulate the distillation knowledge set $\mathcal{D}_k$ with pairs of states and labels:

$$\mathcal{D}_k = \{(s_t^i, Q_{\theta_k}(s_t^i), (s, i) \sim \bar{\mathcal{S}}_k\}. \tag{6}$$

Then $EBS_k$ learns the knowledge shared by its partners. The reference partner list is $RL_k$, and the distillation data set $\mathcal{X}_k = \{\mathcal{D}_p, p \in RL_k\}$. After collecting $\mathcal{X}_k$, $EBS_k$ calculates losses with pairs of states and labels, denoted as (x,y) in Eq.7, from partner EBSs on $Q_{\theta_k}$. The loss of the sample from the partner $EBS_i$ on $EBS_k$ is as follows:

$$L^{ki} = \frac{1}{\|\mathcal{D}_i\|} \sum_{x,y \sim \mathcal{D}_i} [Q_{\theta_k}(x) - y]^2. \tag{7}$$

Then the loss value is weighted by reference weight of $EBS_k$ to its partners and averaged to obtain the final loss value:

$$L_{kd} = \frac{1}{\|\mathcal{X}_k\|} \sum_{\mathcal{D}_i \in \mathcal{X}_k} \left( e^{\varphi_{ki}^l} / \sum_j e^{\varphi_{kj}^l} \right) L^{ki}, \tag{8}$$

where $\varphi_{kj}^l$ represents the reference weight of $EBS_k$ to $EBS_i$ in time slice $l$. The weighted loss value guides the updating of the model parameters:

$$\theta_k \leftarrow \theta_k - \alpha \nabla_{\theta_k} L_{kd}. \tag{9}$$

## 3.2 DDPG-BASED WEIGHT LEARNING

According to Fig. 1(c), the trend and proportion of the same content in different domains is generally inconsistent, which may affect the timeliness and validity of knowledge. Hence, each EBS needs to dynamically maintain its reference weight to partner EBSs to obtain more seasonable and useful knowledge. Consequently, we employ the DDPG algorithm for EBSs to dynamically adjust reference weights to EBSs in the reference partner list.

**Definitions** We define $Ks_t = \{L^{kj}\}, j \in RL_k$ as the state observed by the reference weight agent, which corresponds to the loss of knowledge sets from partner EBSs. The reward is the number of cache hits between distillations, denoted as $r_t^{kd}$. The action of reference weight agent is to calculate the weighted loss of partner EBSs, and thus the parameter of the action network $\sigma_{\vartheta_k}$ is defined by the reference weights to EBS partners:

$$a_t^{kd} \sim \sigma_{\vartheta_k}(a|Ks_t). \tag{10}$$

When the agent take next action, it will observe the state $Ks_t^{'}$ simultaneously. Subsequently, we get distillation tuple $(Ks_t, a_t^{kd}, r_t^{kd}, Ks_t^{'})$, and save it in the distillation replay buffer $M_{kd}$. At regular intervals, reference weight agent on EBSs samples form buffer $M_{kd}$ and gets $\mathcal{Z} = (Ks_t, a_t^{kd}, r_t^{kd}, Ks_t^{'})$, which is abbreviated as $\mathcal{Z} = (s, a, r, s^{'})$ in the following equation. After sampling, we construct the critic network $Q_{\zeta_k}$ to evaluate the utility of actor network and maximize the expected discounted return:

$$J(\sigma_{\vartheta_k}) = E_{\sigma_{\vartheta_k}}[\sum_{(s,a,r,s^{'})\in\mathcal{Z}} Q_{\zeta_k}(s,a)]. \tag{11}$$

The loss can be calculated with the sampled tuples:

$$L_w(\zeta_k) = \frac{1}{\|\mathcal{Z}\|}\sum_{(s,a,r,s^{'})\in\mathcal{Z}} [Q_{\zeta_k}(s,a) - y]^2, \tag{12}$$

where $y = r + \eta Q_{\bar{\zeta}_k}(s^{'}, \sigma_{\bar{\vartheta}_k}(s^{'}))$ is expected discount reward, and $\eta$ is discount factor. The parameters of the target network are denoted as $\bar{\zeta}_k$ and $\bar{\vartheta}_k$, which are initialized with the same parameter as $Q_{\zeta_k}$ and $\vartheta_k$, and updated by soft moving average. Following that, we update the parameters of the critic network:

$$\zeta_k \leftarrow \zeta_k - \alpha\nabla_{\zeta_k}L_w(\zeta_k). \tag{13}$$

Next, we calculate the gradient of the actor network:

$$\nabla_{\vartheta_k}J \approx -\frac{1}{\|\mathcal{Z}\|}\sum_{(s,a,r,s^{'})\in\mathcal{Z}} \nabla_a Q_{\zeta_k}(s,a)\nabla_{\vartheta_k}\sigma_{\vartheta_k}(s). \tag{14}$$

Afterward, we update the parameters of the actor network using its gradient:

$$\vartheta_k \leftarrow \vartheta_k - \alpha\nabla_{\vartheta_k}J. \tag{15}$$

Finally, we update the target network using soft moving average:

$$\begin{aligned}\bar{\vartheta} &\leftarrow \tau\vartheta + (1-\tau)\bar{\vartheta}, \\ \bar{\zeta} &\leftarrow \tau\zeta + (1-\tau)\bar{\zeta}.\end{aligned} \tag{16}$$

### 3.3 CONTENT-AWARE REQUEST ROUTING MECHANISM

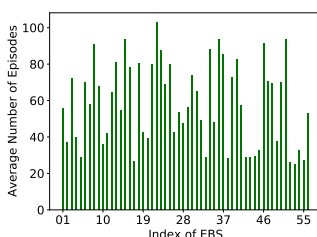

Figure 5: The average episode number of reference weights to all adverse partners dropping below $10^{-3}$. We first calculate the average loss of all partners knowledge in the first 10 episodes. Then we defined partners with greater loss than current EBS as adverse partners.

As illustrated in Fig. 5, we let EBSs refer to all the other EBSs and calculate the average episode number of reference weights to all adverse partners dropping below $10^{-3}$. In each episode, EBS updates the parameter of network with the DDPG-based algorithm. The results indicate that EBS requires an average of 58.3 episodes to dilute the impact of adverse partners, which may slow down the efficiency of model training and reduce CHR performance. Based on the insights from data measurement, we believe that EBSs partners should be dynamic rather than fixed, i.e., EBS need to dynamically select suitable partners for knowledge reference rather than adjust the weights after selecting fixed partners. Motivated by the result in Fig. 3. We defined time slice $T_s$ as the basic unit of time to learn and update the list of reference partner EBSs and maintain reference weights for the EBSs in the list, respectively. The obtained list and weight are only valid within the current time slice. When EBSs move to the next time slice, they learn new lists and weights. In order to observe the similarity shift of video requests over a relatively long period of time, we carried out a more extensive measurement. According to our measurement of daily content similarity in 14 consecutive days, we found two distinct features, as marked in Fig. 2. To sum up, requested contents follow a certain periodicity on a weekly basis, which inspires us to use one week as the strategy cycle $T_p$, and divide the time of one week into time slices of equal length. The device learns the collaboration partners and reference weight respectively in each time slice, and use them at the same time in next week.

The popularity of video content is constantly changing, so we introduce an update cycle, denoted as $T_u$, to reset partner lists and reference weights after several strategy cycles. This approach helps prevent overfitting. Within the same experimental configuration, the length relationship among the three cycles are $T_u > T_p > T_s$. Each strategy cycle consists of equal-length time slices, and each update cycle comprises equal-length strategy cycles. We define $N_p = T_u/T_p$, $N_s = T_p/T_s$. For $EBS_i$ in strategy cycle, we initialize reference partner list as $RL_i = \{RL_i^1, RL_i^2, \cdots, RL_i^{N_s}\}$ and reference weight as $WL_i = \{WL_i^1, WL_i^2, \cdots, WL_i^{N_s}\}$. For the $l$-th time slice in strategy cycle, $RL_i^l = \{EBS_{i1}^l, \cdots EBS_{in}^l\}$, $WL_i^l = \{\varphi_{i1}^l, \cdots \varphi_{in}^l\}$, where $n$ represents partner number of $EBS_i$.

**Partner list adjustment** For each EBS, if the average loss of the distillation knowledge set from other EBSs within the current time slice is lower than the average loss of the training sampling set $B$ on that EBS, it will be removed from the reference partner list. It should be noted that, according to the above process, in each update cycle, the EBS only learns the reference partner list in the first strategy cycle and uses it at the same time in subsequent strategy cycle, rather than dynamically updating the list in the whole process. That is because if the list is updated and iterated throughout each update cycle, it would not only bring about large time and communication overhead, but also affect the efficiency of each EBS in determining the reference weight to partners.

## 4 EVALUATION

In this section, we conduct a series of experiments to evaluate the performance of our framework. Through these experiments, we aim to showcase the superior performance of our framework compared to other existing solutions in real-world scenarios.

**Evaluation setup** We use the real-world data set from Kuaishou company, including 0.35 billion content requests in Beijing for four weeks, and each request consists of a timestamp, location, and content ID. In our experiments, we select the requests from a 14km × 16km area in northwest Beijing (approximately 0.163 billion requests to 2.887 million videos), which spans five central urban districts, including Haidian, Changping, Dongcheng, Xicheng, and Chaoyang Districts. Since the width of the partition does not affect the characteristics of the request (As shown in Fig. 1(c)), we set the width of each domain to 2km and divide this area into 8×7 grids and assume one EBS serves one gridWang et al. (2020). The request amount of each domain ranges from 309,613 to 9,509,393 and 12 domains pose requests above midrange(4,909,503). We sort 56 EBSs in the experiment by request volume from small to large and set new indexes for them. We arrange these EBSs into four equally sized groups based on their average request volume, with group Q1 having the lowest average request volume and group Q4 having the highest.

**Architectures and training parameters** We use PyTorch to develop the experiment and implement our model on a server with fourfold NVIDIA GeForce RTX 4090 GPU cards and Intel(R) Xeon(R) Gold 6230 CPU. The experiment duration is about one and a half hour. We set cache capacity $C$ as 32, discount factor $\gamma$ and $\eta$ as 0.99. $\lambda$ and $\tau$ as 0.25. The popularity network $Q_\theta$ consists of 16 input nodes and one output node, we set up three hidden layers of fully-connected units with 256, 128, and 32 neurons, respectively. The critic network $Q_\zeta$ has $N_{EBS} - 1$ input nodes, one output node, two hidden layers of fully-connected units with 32 neurons each. The learning rates for the popularity network $Q_\theta$, actor network $\sigma_\vartheta$, and critic network $Q_\zeta$ are set to $3.5 \times 10^{-4}$, $4.5 \times 10^{-4}$, and $1.8 \times 10^{-4}$, respectively. Furthermore, we set $T_k$ as 2s, $T_{kd}$ as 10s, $T_r$ as 20s, $T_s$ as 180s, $T_p$ as 168h and $T_u$ as 672h for every EBS. We set the time period $\omega_n$ as $n \times T_s$, and each cache slot has a state dimension of 16×1.

**Baselines** We compare our proposed framework with five baselines as follows. 1) Least recently used(LRU): This method selects and eliminates the most recently unused content. 2) Least frequently used(LFU): This method replaces the content with the lowest reference frequency. 3) Multi-Agent Actor-Critic Algorithm (MAAC) Zhong et al. (2020a): This approach leverages Multi-Agent DRL strategy for cache replacement, but does not involve collaboration between EBSs. 4) MacoCache Wang et al. (2020): This method designs a multi-agent DRL-based collaborative cache strategy, but it needs to specify the partners of each EBS in advance. 5) KDCdCEC-$\alpha$: Our proposed framework before content specificity-based optimization. 6)KDCdCEC-$\beta$: Our proposed framework with content-specificity-based optimization in Section B.

**Evaluation metrics** 1) EBS CHR: As we defined in preliminary, this metric represents the proportion of hit requests to all user requests, indicating the effectiveness of caching. 2)Average latency: This metric refers to the average transmission latency from the moment a user requests a video to the moment it is received. It is a crucial component of a user's quality of experience (QoE) and reflects the efficiency of content delivery. 3)Average traffic cost: Traffic cost refers to the amount of data and computational resources required to exchange information between EBSs and CDN servers. We consider the average traffic cost of each user request as an essential metric for evaluating the resource utilization. Since obtaining the real latency and traffic cost is not feasible, we estimate the average latency by assuming that fetching contents from the CDN server takes ten times longer than retrieving requested contents from the local EBS cache in the case of a cache hitZhong et al. (2020a). Similarly, the average traffic cost between EBSs and the CDN server is assumed to be ten times the average cost between EBSs themselves, providing an approximation of the resource consumption in the systemKrajsa & Fojtova (2011).

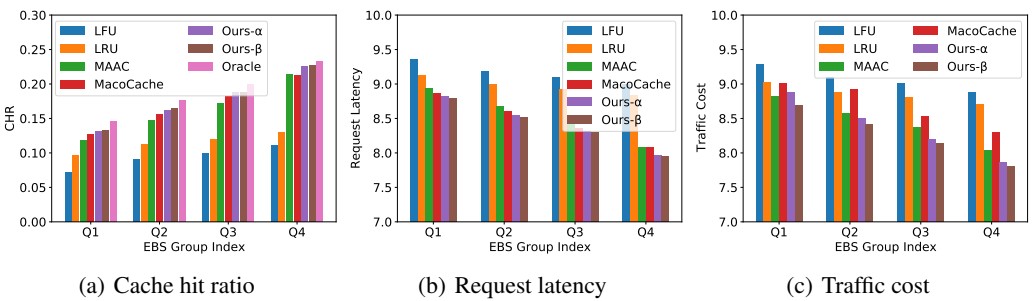

(a) Cache hit ratio  (b) Request latency  (c) Traffic cost

Figure 6: The average video CHR, request latency, and traffic cost of different baseline approaches.

**Overall performance** We measure oracle results under the current experimental conditions. Specifically, we assume that all future user requests are known, and all EBSs adjust the cached contents based on future requests to achieve the maximum CHR. The average CHR and oracle results for each group are shown in Fig. 6(a). KDCdCEC-$\beta$ demonstrated an improvement of 4.71% and 8.05% in CHR compared to the state-of-the-art multi-agent DRL-based approach and the multi-agent collaborative DRL-based approach, respectively. In comparison, the rule-based cache replacement methods LFU and LRU could only achieve 52.27% and 64.30% of the CHR achieved by KDCdCEC-$\beta$. Our focus was primarily on comparing the performances of KDCdCEC-$\beta$, MAAC, and MacoCache in each EBS group. Compared to MAAC, demonstrated an increase of 10.25%, 9.26%, 8.21%, and 5.64% in CHR for Q1, Q2, Q3, and Q4, respectively, indicating that our framework performs better in domains with sparse requests. As for MacoCache, KDCdCEC-$\beta$ exhibited improvements of 4.74%, 4.98%, 2.92%, and 6.03% in CHR for groups Q1 to Q4, demonstrating that our framework enables more precise identification of collaborative partners, particularly for request-sparse EBS. It should be noted that the strategy of collaborative caching is less beneficial for request-dense EBS; however, KDCdCEC-$\beta$ showed the most significant improvement among these EBSs compared to MacoCache. This suggests that the MacoCache approach, which forcibly binds cooperative learning partners for request-dense EBS, hinders the model learning process. Moreover, as illustrated in Fig. 6(b), credit to the excellent caching performance, KDCdCEC-$\beta$ can reduce 8.35%, 6.38%, 1.60%, 0.94% transmission latency compared with LFU, LRU, MAAC, and MacoCache, which improves the smoothness of user experience. The superiority of our approach increased with the request density of each EBS, demonstrating that KDCdCEC-$\beta$ exhibits excellent scalability in complex and heterogeneous edge caching environments.

Table 1: Performance metrics of different baselines

| Method | CHR | Latency | Traffic cost |
|---|---|---|---|
| **KDCdCEC-$\beta$** | **0.1781** | **8.3970** | **8.2661** |
| KDCdCEC-$\alpha$ | 0.1766 | 8.4104 | 8.3608 |
| LFU | 0.0931 | 9.1621 | 9.0691 |
| LRU | 0.1145 | 8.9693 | 8.8548 |
| MAAC | 0.1629 | 8.5338 | 8.4521 |
| MacoCache | 0.1692 | 8.4771 | 8.6942 |

Besides, communication between EBSs incurs additional traffic costs, while improving the cache replacement strategy can reduce the traffic cost between EBSs and CDN servers. Although KDCdCEC-$\beta$ depends on collaboration and timely communication to achieve an optimal cache replacement strategy, it manages to reduce 8.85%, 6.65%, 2.20%, 4.92% traffic cost compared with LFU, LRU, MAAC, and MacoCache, as shown in Fig. 6(c). It is worth mentioning that MacoCache not only transfers video content but also share strategy between EBSs, which leads to more traffic cost than MAAC and KDCdCEC-$\beta$. This result also indicate that our framework can transmit richer information with less traffic cost, which is attributed to the efficient organization of collaboration. Table 1 shows the metric statistics of KDCdCEC and several baselines.

## 5 RELATED WORKS

**Collaborative Edge Cache** In the scenario of edge caching, collaborative strategy between EBSs can improve the efficiency of transmission and caching, thus reducing the transmission delay of content and improving the hit rate of cache Poulimeneas et al. (2016); Nomikos et al. (2021). For better collaborative content sharing, Sung et al. Sung et al. (2016) employed multi-agent Q-learning and combined LFU and LRU to train composite models. Existing edge collaboration strategies are mostly based on content-based collaboration Wu et al. (2018), that is, edge devices acquire the cached content of other devices with certain communication overhead. If the cache fails to hit, other devices will be relied on for a relatively rapid response. However, coordination among cache devices at the strategic level is relatively unexplored, i.e., EBSs share caching intelligence (e.g. parameters of cache replacement model), which is the key focus of our work.

**Online Knowledge Distillation** Our work is more inclined to use the online knowledge distillation method without the pre-trained teacher model. Through learning from the peer prediction to train a group of student models can remit the problem of missing teacher. Zhang et al. Zhang et al. (2018) use individual networks, each corresponding to a student model. Song and Chai Song & Chai (2018) require all student models to share the same early block to further reduce the cost of training. Some works define teacher work in advance Anil et al. (2018); Furlanello et al. (2018); Kao et al. (2021); Hao et al. (2021); Liu et al. (2020). Gao et al. Gao et al. (2021) introduced an adaptation framework for MARL, which focuses on accelerating the training phase by knowledge sharing. However, simply treating each peer as equally important or defining teacher in advance are not robust enough, because the unreliability of teacher selection will limit students' learning to a certain extent. Therefore, there is still a lot of room for improvement in collaborative edge caching.

## 6 FURTHER DISCUSSION

When constructing video popularity features, limited to the given data set, we only use the statistical results of the number of requests in different lengths of time to construct the features of each video, which is a forced simplification. According to our listed related work in the appendix, there are many successful methods for predicting content popularity, and these methods have achieved good results, but this is not our focus of this paper. In addition, the vast majority of video content in our dataset is short video, which may show very different distribution and request characteristics from long video content provided by many other video platforms, such as YouTube. We hope that future work can make a more comprehensive attempt to target these two different video consumption types.

## 7 CONCLUSION

In this work, we commence by analyzing real measurement data, and identify that the inadequate performance of current edge caching is primarily attributed to request-sparse EBSs. To address this problem, we propose using knowledge distillation (KD) to transfer knowledge from request-dense EBSs to request-sparse ones. However, our measurement shows that video requests are susceptible to spatio-temporal drift. Consequently, we design a dynamically adaptive collaboration framework called KDCdCEC, which is incorporates a proficient slot-wise RL agent capable of concurrently managing EBSs with varying storage capacities.simultaneously. Our experiments validate the effectiveness of our algorithm, as we achieve the highest caching hit rate while maintaining low latency and communication overhead.

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

## A NOTIONS

The notions in the paper are shown in Table 2.

Table 2: Notations

| Symbol | Description |
|---|---|
| $C$ | Cache capacity of EBS |
| $N_{EBS}$ | Number of EBS |
| $S$ | Complete edge caching environment |
| $T$ | Time period of operations |
| $D$ | Replay buffer |
| $M_{kd}$ | Knowledge distillation replay buffer |
| $l/p/u$ | Serial number of time slice/strategy cycle/update cycle |
| $RL_k^l$ | Partner EBSs list for EBS$_k$ in time slice $l$ |
| $WL_k^l$ | Reference weight of EBS$_k$ to $RL_k^l$ |
| $s_t^i/a_t^i/r_t^i$ | State/action/reward of cache unit $i$ at time $t$ |
| $\omega$ | Time window for feature abstraction |
| $F$ | Number of time window |
| $\mathcal{B}/\mathcal{S}_k/\mathcal{Z}$ | Sample set from replay buffers$t$ |
| $a_t^{kd}/r_t^{kd}$ | Knowledge distillation action/reward at time $t$ |
| $Ks_t$ | Loss set of all partner EBSs at time $t$ |
| $Ls_k^l$ | Average loss set of $RL_k^l$ at EBS$k$ in time slice $l$ |

# B    OPTIMIZATION WITH PROBABILISTIC SAMPLING STRATEGY

Inspired by the preliminary experimental results of KDCdCEC, we propose an optimization strategy based on request specificity for TDOCoD in KDCdCEC, which can help EBSs obtain more efficient knowledge sets. We evaluate request specificity of each EBS with the content specificity ratio ($CSR$) indicator, which is defined as:

$$CSR_k^l = 1 - \frac{1}{\|N_{EBS}\| - 1} \sum_{i \neq k} \frac{\sum sc_{ki} \in R_k^l}{\|R_k^l\|}, \tag{17}$$

where $CSR_k^l$ represents all requests to $EBS_k$ in time slice $l$. $sc_{ki}$ is the number of same content exist in both $R_k^l$ and $R_i^l$. EBSs with higher $CSR$ have more unique contents in current time slice. Intuitively, it is harder for EBSs with higher $CSR$ to benefit from the knowledge of other EBSs, as knowledge related to less common local content may become ineffective noise in local network training. In such cases, our approach may lead to worse performance on these EBSs. To further evaluate our hypothesis, we define four collaboration formats: 1) Dynamic-collaborative(our proposed strategy), 2) All-collaborative: collaborate with all other EBSs, 3) Random-collaborative: collaborate with randomly assigned EBSs, 4) Non-collaborative: cooperate with no EBS. Then we observe EBS performance under different collaborative formats and find that 1.79%, 3.57%, 71.42%, 17.86% of EBSs performs best under the Non-collaborative, Random-collaborative, Dynamic-collaborative and All-collaborative formats, respectively. Additionally, 5.36% of EBSs perform similarly in all collaboration formats. These results demonstrate that not all EBSs are suitable for collaboration with other devices, inspiring us to optimize the dynamic collaboration process to improve the CHR of more EBSs. As a consequence, we define $P_{rej} = CSR$ as the probability that each EBS rejects requests from other devices, that is, each EBS does not accept all knowledge from partners, but acquires knowledge with a certain probability. Devices with higher $CSR$ have less knowledge from other devices, and vice versa.

# C    ADDITIONAL RELATED WORKS

## C.1    MOBILE EDGE CACHING

Mobile edge caching is a new cache mode emerging in recent years, allowing users to access video content from nearby edge facilities (e.g. EBS), thus reducing traffic cost and content access delay Wang et al. (2014); Ma et al. (2017b). Various methods have been proposed to address different caching situations. Some of them rely on specialized models that assume the video content popularity is known Hachem et al. (2015); Shanmugam et al. (2013); Khreishah & Chakareski (2015),

while others predict the overall content popularity based on characteristics such as historical behavior Niu et al. (2011) and social networks Li et al. (2013). However, predicting content popularity is challenging due to the non-stationary nature of content requests. Existing approaches still lack sufficient adaptability to the highly dynamic and heterogeneous edge cache environment. Several centralized learning-based methods have been proposed Xu et al. (2014); Yu et al. (2018); Zhu et al. (2018b); Zhong et al. (2018); Zhu et al. (2018a); Zhou et al. (2023). Xu et al. Xu et al. (2014) used incremental learning to select the contents of cache, and this method made continuous prediction of input samples. If the pattern of user requests changes, the model is updated so that it can accurately predict user requests. In addition, other works based on multi-armed bandit model Blasco & Gündüz (2015), Q-learning Sadeghi et al. (2017) and transfer learning Baştuğ et al. (2015) have been proposed. However, these methods face transmission delays and high training costs in edge scenarios, making large-scale deployment challenging.

## C.2 DEEP REINFORCEMENT LEARNING

Reinforcement learning has been widely used as an efficient tool to solve kinds of sequential problems that can be modeled as MDP. Its application enables EBS to optimize cache strategies through the interaction with edge environment Ye et al. (2021); Lei et al. (2017); Somuyiwa et al. (2018). With the advent of deep neural networks, EBS can learn low-dimensional representations using high-dimensional raw data. Recent works have applied DRL methods and combined them with practical problems. For example, Zhong et al. Zhong et al. (2020b) used the DRL model based on KNN, and Wu et al. Wu et al. (2019) introduced the idea of LSTM into the DRL, which expanded a new idea for the algorithm design in our work. To enable EBSs to better manage and train cache autonomously, Zhong et al. Zhong et al. (2018) proposed a DRL-based content cache framework, which does not require a prior content popularity distribution. In addition, multi-agent DRL extends the application of DRL to collaborative or competitive scenarios Zheng et al. (2018). No matter how DRL algorithm is designed, an effective model cannot be separated from sufficient and abundant training data. However, in edge scenario, the model with sparse training data may become the bottleneck to improve the efficiency of the entire edge cache system.

## D OVERVIEW OF KDCDCEC

RL agents are deployed on each EBS at the network edge, where they estimate the popularity of contents and manage cached videos while serving user requests. The framework of KDCdCEC is illustrated in Fig. 7. To address the temporal drift of user requests, we have designed a KD-enhanced DRL-based cache replacement strategy, which forms the foundation of KDCdCEC. As the patterns of user preferences changing vary across domains, KDCdCEC incorporates a DDPG-based weight learning method that updates the reference weights of EBSs to their partners using spatial awareness. Furthermore, considering that EBSs in certain domains may face challenges in accessing reliable partners, KDCdCEC is equipped with a content-aware request routing mechanism strategy to regulate the collaborative cycle of the KD strategy on EBSs.

## E EXPERIMENTS

### E.1 ABLATION STUDY ON COLLABORATIVE STRATEGY

To understand the impact of each component in KDCdCEC, we conduct ablation studies to demonstrate the following: 1): The superiority of collaborative caching with KD. 2): Dynamic collaborative caching contributes to overall performance. 3): Content specificity-based optimization is helpful.

Apart from the previously mentioned KDCdCEC-$\beta$ and KDCdCEC-$\alpha$, we also include the following baselines: 1)Non-collaborative: EBSs train their DRL-based cache replacement models with no knowledge from other EBSs. 2)Random-collaborative: EBSs train their DRL-based cache replacement models with knowledge from randomly selected EBSs. We set the partner number as 8. 3)All-collaborative: EBSs train their DRL-based cache replacement models with knowledge from all other EBSs. The metric statistics of KDCdCEC and several collaborative approaches are presented in Table 3.

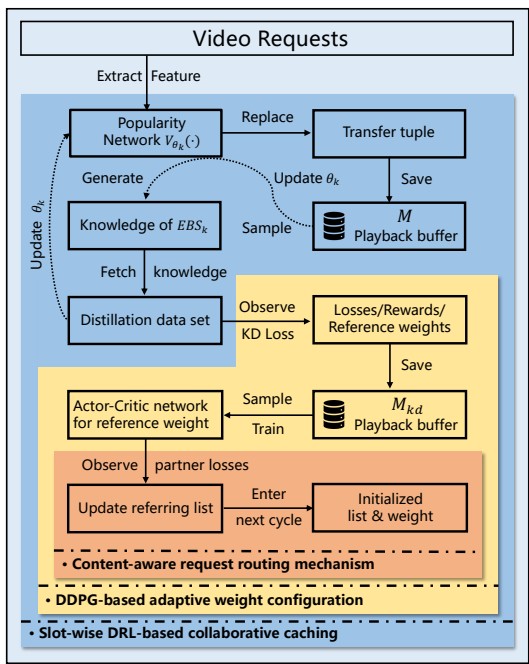

Figure 7: The framework of KDCdCEC.

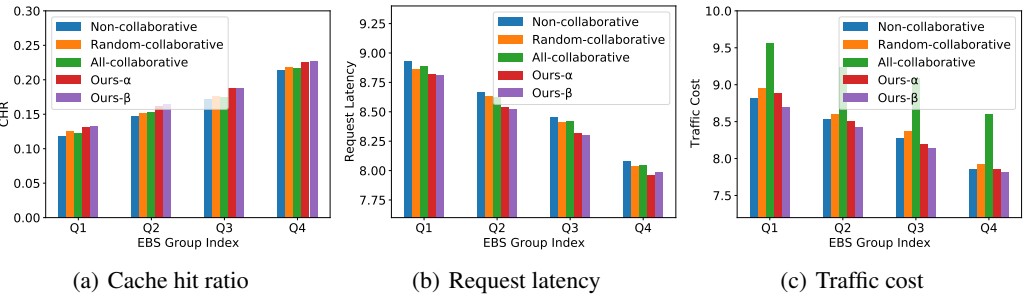

(a) Cache hit ratio       (b) Request latency       (c) Traffic cost

Figure 8: The average video CHR, request latency, and traffic cost of different approaches for different collaborative strategies.

Table 3: Performance metrics of different collaborative approaches

| Method | CHR | Latency | Traffic cost |
|---|---|---|---|
| **KDCdCEC-$\beta$** | **0.1781** | **8.3970** | **8.2661** |
| KDCdCEC-$\alpha$ | 0.1766 | 8.4104 | 8.3608 |
| Non-collaborative | 0.1629 | 8.5338 | 8.3709 |
| Random-collaborative | 0.1680 | 8.4880 | 8.4587 |
| All-collaborative | 0.1671 | 8.4959 | 9.1232 |

**Superiority of collaborative caching with knowledge distillation**: As shown in Fig. 8(a)&(b), all four approaches relying on the KD-based collaborative caching strategy outperformed the Non-collaborative strategy in terms of CHR. This demonstrates that introducing knowledge distillation to the edge caching problem enhances the caching intelligence of EBSs. Specifically, KDCdCEC-$\beta$, KDCdCEC-$\alpha$, Random-collaborative and All-collaborative can increase average CHR of 9.33%, 8.41%, 2.59%, 3.12% compared to Non-collaborative. Furthermore, they reduced latency by 1.60%,

1.45%, 0.44%, 0.54%, respectively. Although using KD incurs additional communication traffic costs between EBSs, cache hits can save traffic costs between EBSs and CDN servers.

**Dynamic collaborative caching contributes to overall performance**: KDCdCEC outperforms among KD-based approach. The average CHR metric of KDCdCEC-$\beta$ is 6.57% and 6.02% higher than All-collaborative and Random-collaborative. Our proposed framework also reduces 0.96% and 1.03% request latency compared to All-collaborative and Random-collaborative. Moreover, dynamic collaborative caching results in prominent fewer traffic costs, as shown in Fig. 8(c). This can be attributed to the fact that the average number of collaborative partners in KDCdCEC is less than the fixed 8 partners in Random-collaborative. EBSs equipped with KDCdCEC achieved better performance with fewer reference partners.

**Content specificity-based optimization is helpful**: We implement content specificity-based optimization in KDCdCEC-$\alpha$ and get KDCdCEC-$\beta$. According to the experimental results, KDCdCEC-$\beta$ outperformed its predecessor in every metric. Although the improvement may not seem significant, we consider this optimization as a practical approach to better identify possibilities for collaboration in solving multi-node collaborative training problems in edge networks.

### E.2 HYPERPARAMETER ANALYSIS

To verify the validity of the three time-related cycles we selected, we further conduct orthogonal experimental design. The result indicates that the degree of influence of the three cycles on CHR is: $T_s \gg T_p > T_u$, where the choice of $T_s$ greatly affects caching performance, while $T_p$ and $T_u$ have very little effect on the results. Moreover, we conducted ANOVA for the above cycles to determine their significance level. Judging by the p-value of each factor, it is more than 95 percent likely that $T_s$ is significantly related to CHR, while the significance of $T_p$ and $T_u$ is not evident. This result supports our conclusions from the orthogonal experimental design. Therefore, in the evaluation section, we will mainly focus on the relationship between $T_s$ and CHR.

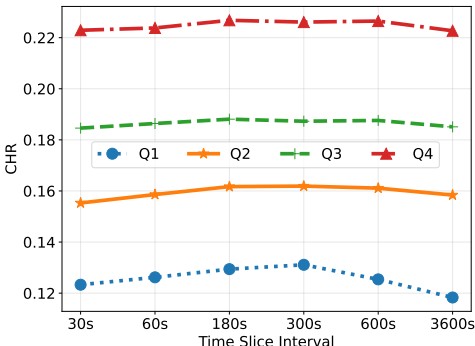

Figure 9: Average CHR with different time slice interval

The time slice interval is the observation period during which EBS makes adjustments to the reference partner list, leading to different forms of collaboration. A narrow time slice may result in a short-sighted strategy, while a wide time slice may lead to an outdated strategy. Therefore, we are bound to find suitable time slice interval for EBSs. As shown in Fig 9, we varied the time slice interval from 30 seconds to 3600 seconds and selected six representative widths to test the performance of EBSs in each group. Group Q1 and Q2 performed best when the time slice interval was set to 300 seconds, while group Q3 and Q4 achieved better performance with a narrower time slice. Compared to group Q3 and Q4, group Q1 and Q2 had much fewer requests for model training, which may have delayed model convergence and necessitated more time for request-sparse EBSs to identify beneficial partners rather than making hasty decisions. These results provide guidance for selecting the time slice interval hyperparameters.

Furthermore, we believe that the experiment with a smaller cache size $C$ can better reflect the utility of the caching algorithm, because the fault tolerance of the EBS cache is greatly reduced when the cache size is small (we set $C$ as 32 in most of our experiments). In order to ensure the rigor of

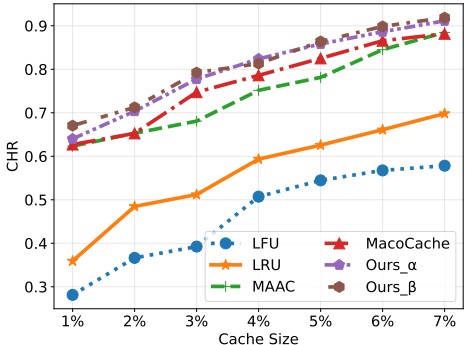

Figure 10: The average CHR of different approaches for different cache size on each EBS.

the experiment, we change the edge cache capacity $C$ from 1% to 7% of the total content number and show the results in Fig. 10. We can find that our approach have higher CHR than MacoCache and MAAC in every cache size conditions, not to mention the simple rule-based approach LFU and LRU.

### E.3 CAPABILITY OF KDCDCEC TO TEMPORAL-DRIFT

To evaluate the performance of KDCdCEC in a temporal-drifting edge environment, we generate two datasets where the content popularity subject to Zipf distributions (Each dataset consists of 2.887 million different videos). In the static-Zipf dataset, content popularity remained stationary, allowing us to evaluate how well KDCdCEC could learn content popularity. In the dynamic-Zipf dataset, the popularity of video content followed the Zipf distribution for a short time and then continuously changed. This dataset was used to evaluate whether KDCdCEC could adapt to changing popularity and how quickly it could do so. We set the Zipf parameter as 0.6 in both synthetic datasets. In addition to the synthetic datasets, we also tested our proposed algorithms using real-world traces.

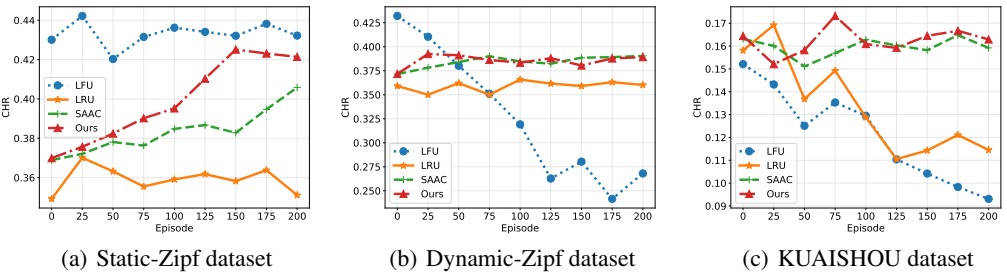

(a) Static-Zipf dataset     (b) Dynamic-Zipf dataset     (c) KUAISHOU dataset

Figure 11: The CHR of different algorithms in single edge cache.

We selected the rule-based algorithms LFU and LRU, as well as SAAC (single-agent version of MAAC), as baselines. During the evaluation, we disabled communication between EBSs and tested the performance of each baseline with a single-agent setting.

As illustrated in Fig. 11(a), in static-Zipf dataset evaluation, LFU achieves the best performance and converges quickly, while LRU gets the worst performance. That supports the tight static regret of LFU and linear static regret of LRU Paschos et al. (2019). KDCdCEC and SAAC achieved similar final performance to LFU, with approximately 1% and 2.5% worse CHR, respectively. This indicates that our proposed algorithm can effectively capture video content popularity. Furthermore, we observed that KDCdCEC converged much faster than SAAC, likely due to the decoupling of cache content and cache space, which reduces the algorithm's complexity and accelerates convergence.

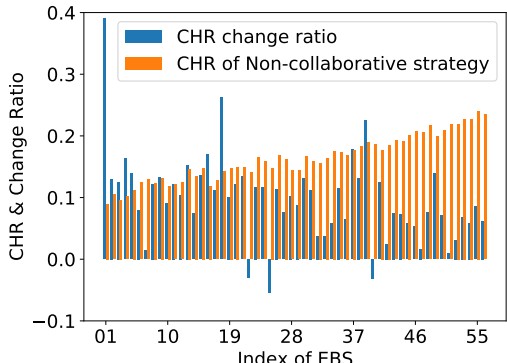

Figure 12: Change ratio of CHR using KDCdCEC compared with Non-collaborative strategy.

When video content popularity are continuously changing, there was a enormous decline in the performance of LFU, as illustrated in Fig. 11(b). However, the performance of LRU does not change significantly because it is insensitive to history requests. However, the performance of LRU remained relatively stable because it is insensitive to historical requests. KDCdCEC achieved a performance similar to SAAC in capturing dynamic content popularity after several episodes. The time window width of KDCdCEC is designed based on insights from large-scale data measurements, allowing it to extract video request features accurately and accelerate model convergence.

Fig. 11(c) presents the evaluation results of the algorithms using the Kuaishou dataset, where the distribution of video requests is more complex than the Zipf datasets mentioned above. In this situation, KDCdCEC outperformed all the other baselines, demonstrating that our method is suitable for temporal-drifting edge environments.

## F REAL-WORLD TRACE INSPECTION

In order to demonstrate the effect of KDCdCEC deployment in an spatiotemporal data-drifting edge scenario and illustrate the dynamic collaboration process among EBSs, we conducted a real-world trace inspection.

The variations in CHR compared to the performance of adopting the Non-collaborative strategy for each domain are shown in Fig. 12. The CHRs of the request-dense domains increase by 6.08% on average, and request-sparse domains improve by 10.82% on average. Since request-sparse domains account for over 78% of the whole domain and respond to 54.8% of the entire requests, we believe that the cache performance improvement in request-sparse domains is the key for KDCdCEC to achieve excellent performance. As illustrated in Fig. 12, we observe a decline in CHR for three EBSs after implementing our framework. We attribute this to the dissimilarity of user requests in these domains compared to others, as well as inconsistent trends in content popularity, which interferes with the training process of the models.

To observe the dynamic collaboration process among EBSs and explain the effect of KDCdCEC using heuristic knowledge, we selected three request-sparse domains for verification. We chose two specific times of the day to observe the collaboration of EBSs deployed in these three domains and their reference weights to their partners.

### F.1 DOMAIN 1

This domain primarily comprises several high-tech companies in Shangdi, such as Kuaishou, Baidu, and Didi. The top 5 reference weights are illustrated in Fig. 13 with orange font. We can observe that at ten o'clock in the morning, the EBS in Shangdi mainly refers to the nearby EBSs, which aligns with the data measurement insight in Wang et al. (2020) that more than 85% of EBSs exhibit content similarity of greater than 65% to neighboring EBSs. At 22:00, the EBS in Shangdi changes its reference strategy to refer more to settlement areas like Huilongguan and Tiantongyuan, which are situated above the selected domain. These domains generate a high volume of video requests due

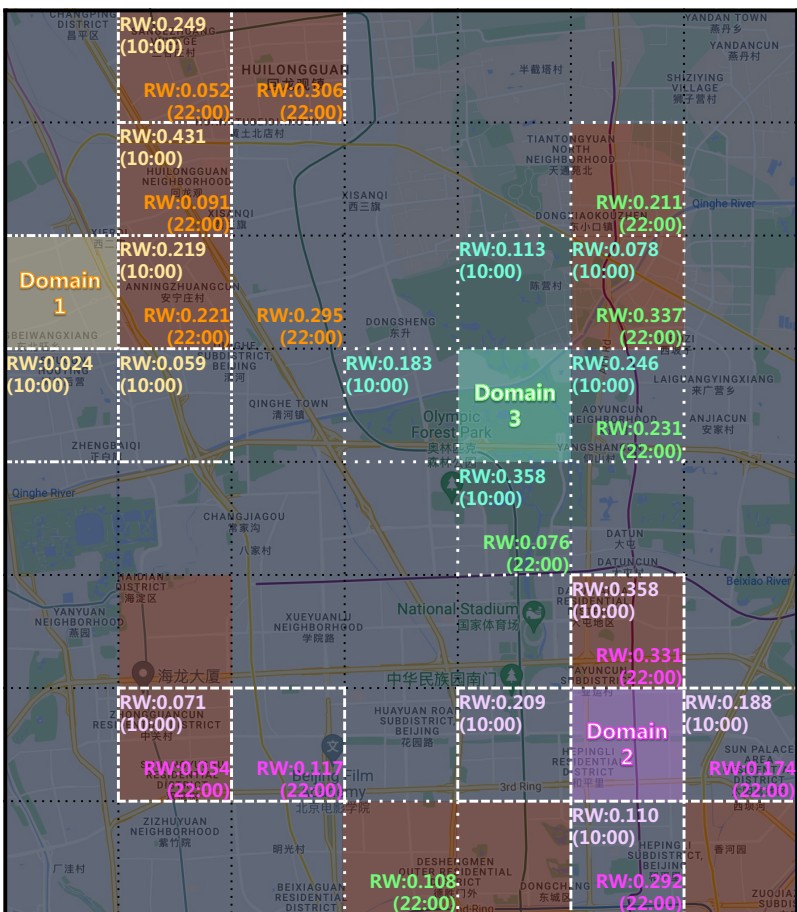

Figure 13: Top 5 reference weights of the selected EBSs to other EBS at 10:00 and 22:00 of the same day. We select 3 representative request-sparse domains in Shangdi high-tech Industrial zone, Chaoyang university cluster and Olympic forest park respectively for verification. The reference weights (RW) of each domain are marked with similar colors and same borders on the map with the corresponding moment. 12 domains shaded in dark red on the map have higher request volume than midrange. The insights from the map will be explained detailedly in the text.

to their dense population, and we assume that a significant number of people working in Shangdi reside in these communities.

## F.2 DOMAIN 2

This domain is primarily composed of universities in Chaoyang District, such as UIBE, BUCT, and BUCM. The reference weights at 10:00 AM and 10:00 PM are illustrated in Fig. 13 with pink font. We observe that the EBS in domain 2 primarily refers to nearby EBSs and EBSs in Haidian District, where numerous universities and hi-tech parks are located. This is likely because university students exhibit similar content preferences. Our proposed algorithm can capture the occupational characteristics of users within the domain, thereby influencing EBS collaboration. It's worth mentioning that nearly 83% of the collaborative partners to the EBS in Tsinghua University in Haidian District are located around universities.

## F.3 DOMAIN 3

This domain mainly covers the Olympic forest park. The reference weights at 10:00 AM and 10:00 PM are illustrated in Fig. 13 with green font. In the morning, the EBS in domain 3 refers to nearby EBSs. However, during the night, two distant EBSs are referenced, one of which is over 8 kilometers

away from the EBS in domain 3. We observe that this EBS refers to three request-dense EBSs, including the aforementioned distant EBSs. We speculate that the EBS in the Olympic Forest Park receives a small quantity of requests at night, and the request characteristics are not prominent. Consequently, more EBSs with a high volume of requests are referenced to assist in its own caching decision, which confirms that domains with a higher request volume are more conducive to training better models, and their knowledge can be leveraged to assist in training request-sparse EBSs.

