# OpenReview forum: "Video Caching at Data-drifting Network Edge: A KD-based Cross-domain Collaborative Solution"
_ICLR.cc/2024/Conference — Submitted to ICLR 2024_

### Official Review · Reviewer_7VV8 · 2023-10-28

**Soundness:** 3 good
**Presentation:** 2 fair
**Contribution:** 2 fair
**Rating:** 5
**Confidence:** 3

**Summary:**

Edge base stations (EBS), a cost-effective algorithm, is a well solution to address network congestion and quality decline caused by explosive growth of video content streaming. To tackle this issue, the authors proposed KDCdCEC, an adaptive knowledge-distillation based cross-domain collaborative edge caching framework to overcome the temporal drift and spatial drift in the caching data simultaneously. The authors fully analyzed and verified the data and experiments, but there is a high degree of redundancy in writing and poor sense of paragraph hierarchy.

**Strengths:**

1. The research topic and method have practical significance and can effectively reduce the network congestion problem.
2. The authors measure and analyze the video requests in the real world, and the experimental results verify the effectiveness of their method.

**Weaknesses:**

1. The methods compared in the articles are all before 2020, so they are not timely and persuasive. It’s better to compare it with the latest methods, such as Edge Caching or KD related.
2. Many of the hyper-parameters are given directly, without explanation and associated hyper-parameter analysis experiments.
3. The readability of the article is not good, and the framework of the overall model may be better explained in the main content.
4. For data analysis, the author proposed temporal drift and spatial drift, and designed different algorithms for them respectively. However, the experimental analysis lacks a stronger explanation for temporal and spatial, or the lack of relevant results indicates whether each part is right.

**Questions:**

How exactly is the data collected and will it be used as a public data set?

---

> ### Author Response · Authors · 2023-11-23
>
> #Response to weakness 1
> ##We also investigated the latest related methods, and found that they are not suitable for migrating to the edge cache scenario we set. Moreover, many methods are not open source, and forced migration will bring us a lot of work and is unfair.
>
> #Response to weakness 2
> ##Due to space limitations, our detailed analysis and results on hyperparameters are presented in the appendix (E.2).
>
> #Response to weakness 3
> ##Your suggestion has been taken and our statement of the problem has been refined In subsection 2.2.
>
> #Response to weakness 4
> ##Due to space limitation, both our ablation study and real-world trace evaluation are presented in the appendix( E.1,E.3,F), and these supplementary experiments are intended to demonstrate the effectiveness of each module of our work.
>
> #Response to weakness 5
> ##We have re-segmented individual chapters and made the core content bold or italic. In addition, we have streamlined the language of some paragraphs to increase fluency.
>
> #Response to weakness 6
> ##The data is collected by Kuaishou Company, but we are not clear about how it is collected. They only provide us with the latitude and longitude of the request, the time of the request and the video ID.

---

### Official Review · Reviewer_eQVV · 2023-10-31

**Soundness:** 3 good
**Presentation:** 2 fair
**Contribution:** 3 good
**Rating:** 6
**Confidence:** 2

**Summary:**

This paper investigates collaborative edge caching in spatiotemporal data-drifting edge scenarios. Through the analysis of real-world measurement data, the authors observed correlations in requests across both spatial and temporal dimensions. They also observed a notable relationship between the volume of requests and the Cache Hit Ratio (CHR). Leveraging these insights, they introduced an adaptive collaboration framework named KDCdCEC. Extensive experiments, including comparisons with existing strategies, show the efficacy of their proposed approach. The paper further provides how KDCdCEC dynamically adapts its caching strategies based on user activity variations and domain-specific characteristics.

**Strengths:**

-	This paper proposes a novel approach to handle cache performance in spatiotemporal data-drifting edge scenarios. The paper’s attention to both request-dense and request-sparse domains provides a more intricate understanding of caching dynamics, differentiating it from conventional generalized solutions.
-	By analyzing real-world measurement data, the authors ensure that their findings and proposed solutions are grounded in actual user behaviors and scenarios. The proposed KDCdCEC framework has been rigorously tested against other strategies, providing a comprehensive view of its efficiency and superiority.
-	Addressing the collaborative edge caching problem in the context of spatiotemporal data drift is highly relevant, especially in an era where data is continuously generated, and its patterns of access constantly evolve. Further, this paper’s findings could have practical implications, potentially influencing how edge caching is approached in various real-world scenarios, from tech hubs to university zones to parks.

**Weaknesses:**

-	In the system model, the state representation includes the number of requests over intervals for C cached videos and the top-C videos selected based on their feature modulus from the uncached pool. This approach might miss out on caching videos that are about to become popular, especially if their current feature modulus isn't among the top-C. An explanation on the rationale for equating the number of uncached videos to the cached ones in the state would be beneficial. Additionally, it would be insightful to provide an explanation on the decision to rely solely on the number of requests as a state.
-	The paper could benefit from a complexity analysis of the proposed framework or comparison with other algorithms. Understanding its computational and memory requirements in various scenarios would be essential for practical deployment in real-world systems.
-	The paper could be enhanced by explicitly discussing its limitations and potential direction for future work. One possibility can be the consideration of alternative features beyond just the number of requests. Providing such discussions would give readers a more comprehensive view and could guide subsequent research in this area.

**Questions:**

-	To improve readability, it might be helpful to  re-locate Figures 1 and 2 to their references in Section 2-1, even though they are currently located in Section 1.
-	In eq. (1), the terms r^i_t and r^j_t are defined as the cumulative numbers of requests to the content in i-th cache slot and j-th candidate video sequence, respectively. It might be clearer to use distinct notations for each, to avoid potential confusion between the cache slot and the candidate video sequence.

---

> ### Author Response · Authors · 2023-11-23
>
> #Response to weakness 1
> ##We calculate the feature modulus of all videos in each feature extraction, which can largely avoid the problem of missing popular content. In the formulation of the Problem, we consider that $C$ contents in the cache should be replaced each time, and in the most extreme case, $C$ new contents should be filled. Then, we carry out slot-wise transformation to decouple the stored content from the storage location, so that EBS with different capacities can be adapted. We acknowledge that relying only on the request number to extract features is a simplification, but the features extracted from the number of requests are straightforward and effective.
>
> #Response to weakness 2
> ##The experiments on the traffic cost of different baselines can reflect the cost of data exchange, which can reflect their complexity to a certain extent (for example, although the Baseline only relying on content-level cooperation can achieve a relatively close cache hit rate to our proposed algorithm, However, it faces a traffic cost much higher than our algorithm), the hardware used for training and the training time have been supplementary explained in the architectures and training parameters in section 4.
>
> #Response to weakness 3
> ##We added the further discussion section before conclusion (Section 6).
>
> #Response to weakness 4
> ##We adopted your suggestion and the layout has been adjusted
>
> #Response to weakness 5
> ##We adopted your suggestion and changed eq.(1) to a clearer form.

---

### Official Review · Reviewer_CiwC · 2023-11-06

**Soundness:** 3 good
**Presentation:** 2 fair
**Contribution:** 3 good
**Rating:** 6
**Confidence:** 3

**Summary:**

The paper provides a DRL video caching mechanism at the edge level with knowledge transfer capabilities among edge base stations (EBS). Using real-dataset, the authors analyzed the data and observed that caching is affected by the sparsity of request distribution (temporal and spatial drifts) at the edge caches, thus KD-based approach is proposed for sharing the requests and drift knowledge among EBSs. Evaluation results show improvement in terms of cache hit rates compared to some approaches, e.g. LRU, LFU, ..etc.

**Strengths:**

- Real dataset is used and some insights from data are provided
- Using RL to model the evicting policy and decides which file to evict.
- Nice problem and good visualization.
- Very detailed evaluation results and a good combination of KD sharing and DRL algorithm.

**Weaknesses:**

- The DRL formulation is not capturing some key factors of streaming systems dynamics including different video file-size distribution and EBS heterogeneities.
- DRL state representation seems to only look at one aspect - the file request in a given past window and does not account for other factors such as its retrieval time (latency), size, popularity ...etc. Size and retrieval time are both important factors and will play a role in transition from one state to another. How the proposed model will change accordingly?
  - For example, one big file needs to replace a few small files and thus the action will change (action will no longer be binary but rather will need to span a few entries in the action matrix and hence the action will be very hard (combinatorial problem) - which video files to evacuate in order to place the new big file)
- Reward seems to be designed to only capture the local reward and no global term. How to ensure consensus in terms of convergence? No convergence analysis is provided.
- Scalability and scaling the algorithm to cover the more general form of different file sizes and servers differences will make the state space and action space (NP Hard, comb. problem) very large and thus very complex to solve.
- Each EBS serves a distinct region (no overlap, as per the paper) and optimizing each EBS individually should be sufficed in my opinion. How sharing among servers will help here?
- How is the three defined temporal metrics related to video file arrival rate?. Why not adopting arrival rate instead since it is widely used and correlate/encapsulate at least two of these three metrics?

**Questions:**

Please check the questions in the weaknesses.

**Details Of Ethics Concerns:**

None!

---

> ### Author Response · Authors · 2023-11-23
>
> #Response to weakness 1
> ##Your suggestion has been adopted and the notation and description have been unified.
>
> #Response to weakness 1
> ##Many similar works do not consider the file size, or treat it as the same size. Our focus is on how to enhance EBS collaboration on a strategy-level, so the size of the video is simplified, and our slot-wise design can be adjusted to almost all different EBS storage space.
>
> #Response to weakness 2
> ##The number of requests in the past time windows is the core metric to evaluate popularity, as the input to the neural network. We follow the assumptions made in previous work and do not regard retrieval time (latency) and file size as the focus of this work.
>
> #Response to weakness 3
> ##The goal of all EBS is to improve their own cache hit rate, and the global goal is also consistent with this. Because in our framework, there is no content-level cooperation between EBS, so each EBS does not need to consider what content is cached by other EBS, and only needs to search for caching strategies to refer from beneficial partners.
>
> #Response to weakness 4
> ##This work makes a slot-wise design, and the method can be used for devices with different cache sizes. If the size of the video and the capacity of EBS are considered, an additional replacement algorithm needs to be designed for implementation, which is not the focus of this work
>
> #Response to weakness 5
> ##Although the regions served by different EBS do not overlap, according to our data measurement results in Fig. 1(a), we find that the request volume of different regions is positively correlated with the cache hit rate when the same model is deployed. We believe that EBS with larger request volume has effective strategies to assist its cache replacement. Moreover, this strategy can assist the EBS devices with sparse requests to perform cache replacement better.
>
> #Response to weakness 6
> ##The second metric is the average latency calculated with CHR, but it gives a more intuitive and significant representation of the latency experienced by the user. The communication overhead of the third metric is negatively correlated with CHR, and positively correlated with the number of information exchange between among EBSs and CDN when using different algorithms, which can comprehensively reflect the performance of the algorithm.

---

### Official Review · Reviewer_V6gK · 2023-11-07

**Soundness:** 2 fair
**Presentation:** 1 poor
**Contribution:** 1 poor
**Rating:** 3
**Confidence:** 4

**Summary:**

Collaborative edge caching can offer a cost-efficient strategy for enhancing video caching and streaming. This paper conducted a measurement study and revealed that the primary factor influencing caching performance is the spatio-temporal diversities in request patterns. To tackle the issue of sparse requests in specific Edge Base Stations (EBSs), the paper proposes to use knowledge distillation to transfer knowledge from request-dense EBSs to request-sparse ones. To further account for the spatio-temporal drift in request distribution across EBSs, the paper designs an adaptive KD-based cross-domain collaborative edge caching framework, named KDCdCEC. The proposed framework consists of three major components: 1) a RL agent to make caching replacement decisions in each EBS; 2) a deep deterministic policy gradient-based algorithm to adaptively configure the reference weights for adaptive KD; 3) a content-aware request routing mechanism. Experiment results shows the effectiveness of KDCdCEC under a specific caching setting.

**Strengths:**

1. In this paper, a novel approach to collaborative edge caching is explored. Rather than concentrating on the sharing of cached content among partners, the paper's primary focus is on the sharing of caching policies among partners. This approach aims to minimize the communication overhead between partners that typically arises from content sharing.
2. The paper applies knowledge distillation to facilitate the training of multi-agent RL in addressing the challenges posed by request heterogeneity and dynamics. The application of  multi-agent RL with knowledge distillation to collaborative caching problem is relatively new.

**Weaknesses:**

1. The problem statement or definition of collaborative edge caching in this paper lacks clarity and support.
1) The paper assumes that the EBSs collaborate by sharing caching policies rather than cached content. This assumption gives rise to two problems. Firstly, considering that each EBS has limited cache storage, if they don't share cached content among themselves, every cache miss would necessitate content retrieval from remote CDN servers. Consequently, the potential benefits of collaborative edge caching are constrained since it doesn't significantly reduce backbone traffic. Secondly, the paper also highlights the spatial drift and diversity of request patterns in different EBS domains. This raises a question about the rationale behind advocating the sharing of caching policies among EBSs. For example, if one EBS serves requests with a LRU pattern, while another EBS caters to a LFU pattern, how would these two EBSs benefit from sharing caching policies when their request patterns differ?
2) This paper studies a caching problem with a fixed number of video contents, which significantly deviates from real caching systems where the number of videos or objects varies over time. Furthermore, the paper assumes the CDN server have cached all the videos. However, considering the substantial number of videos (i.e., 2.887 million) mentioned in the evaluation section, this assumption might be overly stringent, as CDNs typically have relatively limited storage capacity.
2. The presentation of the proposed framework in the paper lacks clarity, and certain claims about the techniques are not well substantiated.
1) The paper claims that the RL agent is slot-wise and can adapt to EBSs with different storage sizes. However, it's noted that the action space of the RL agent is C \times C, where C seems a fixed number across all EBSs throughout the paper (including the evaluation section). If C does indeed differ among EBSs, then during knowledge distillation, how do you handle the variable input state size, which is 2C \times F?
2) What is the content-aware request routing mechanism? I did not find any technique description about it.
3) The paper uses the names and notations interchangeably, which leads to confusion regarding the number of neural networks within the proposed framework, their specific functions, and how they are trained, particularly in the case of the popularity network.
4) What is the ration behind the design of the reward in Equation 1? As the reward is determined by consecutive cache misses to the same cache slot, if that cache slot stores a popular video, the delay of reward for the popular video is much longer than the delay of reward in the situation of storing an unpopular video at the slot. This further indicates that the training data (i..e, the transition tuple (s, a, \tao, s’)) can be potentially imbalanced.
5) How does the reference weight agent adapt to dynamic partner lists? If two EBSs are no longer pattern, it seems the corresponding action network parameter in the reference weight agent should be zero. Then, how do you enforce this when training the reference weight agent?
6) it is unclear how the proposed framework update reference partner list.
7) The training location and process of the RL agents and neural networks are not detailed in the paper, and additional information is not provided. For example, what is the training convergence speed and time of the RL agents? What is the size of training data for different EBS domain. This is important as the edge server often has limited computation and storage resources.
3. The paper's evaluation results are not sufficiently robust or convincing. What is the reason to choose a cache size of 32? The caching size should be related to the total active video content number. Besides, the proposed framework should be evaluated on various caching size setting.
4. The related work section misses literatures about multi-agent RL and multi-agent RL with knowledge distillation. Consider the following examples:
Leonardos, Stefanos, et al. "Addressing Out-Of-Distribution Joint Actions in Offline Multi-Agent RL via Alternating Stationary Distribution Correction Estimation." Advances in Neural Information Processing Systems 36 (NeurIPS 2023). 2023.
Tseng, Wei-Cheng, et al. "Offline Multi-Agent Reinforcement Learning with Knowledge Distillation." Advances in Neural Information Processing Systems 35 (2022): 226-237.
Gao, Zijian, et al. "KnowSR: Knowledge Sharing among Homogeneous Agents in Multi-agent Reinforcement Learning." arXiv preprint arXiv:2105.11611 (2021).

**Questions:**

See questions in the Weakness section.

---

> ### Author Response · Authors · 2023-11-23
>
> #Response to weakness 1 and  4
> ##Your suggestion has been taken and our statement of the problem has been refined In subsection 2.2.
>
> #Response to weakness 2
> ##The increase of cache hit rate can reduce the traffic of the backbone network, and the focus of this paper is to optimize the strategy-level cooperation. As for the situation you mentioned that the request patterns of different regions are different, in the framework we designed, each EBS can adjust the appropriate cooperative partner according to the gain and loss of referring to the caching strategy of EBS in different regions.
>
> #Response to weakness 3
> ##As we introduced in Section 3.1, we transform the problem into a slot-wise approach that can adapt to different cache sizes. According to our measurements, videos and their average requests show a long-tail distribution, and about 80% of video content has less than or equal to one request per day. Through simple filtering and screening, popular content can be stored in CDN to meet the requests of the vast majority of users, so as to effectively alleviate the storage pressure.
>
> #Response to weakness 5
> ##During the formulation of the Problem, we consider replacing C contents in the cache each action, so the initial dimension of the state space is $C \times F$. After we perform the slot-wise transformation, when cache misses, the features of the content in each slot are input into the popularity network for scoring, that is, the dimension of the network input vector is $1 \times F$, and then the content with the lowest score is selected for replacement.
>
> #Response to weakness 6 and 7
> ##Your suggestion has been adopted and the notation and description have been unified.
>
> #Response to weakness 8
> ##We refined the presentation of the reward design section to reduce ambiguity. Delayed rewards in reinforcement learning refer to the situation where the consequence, or reward,  of an action is not immediately experienced but occurs after some delay.  In other words,  when an agent takes an action, it may not receive feedback or a reward immediately,  and there could be a temporal gap between the action and the corresponding outcome.  However, in the context of our problem, the RL Agent must have a corresponding reward after each action, and the problem of Delayed rewards may not exist here.
>
> #Response to weakness 9
> ##The parameter of actor network of reference weight agent is the reference weight of EBS to each member in its partner list. The EBS not in partner list will not be referenced, and its reference weight will not be included in the actor network. The input dimension of actor network will be adjusted accordingly in this time slice.
>
> #Response to weakness 10
> ##The content of the adjustment of the partner list is introduced in the last paragraph of Section 3.3, and we have made additional annotations for your review.
>
> #Response to weakness 11
> ##The parameter of actor network of reference weight agent is the reference weight of EBS to each member in its partner list. The EBS not in partner list will not be referenced, and its reference weight will not be included in the actor network. The input dimension of actor network will be adjusted accordingly in this time slice.
>
> #Response to weakness 12
> ##Experiments with different baseline performance versus cache size were moved to the appendix (E.2) due to space limitations.
>
> #Response to weakness 13
> ##New related work has been added to section 5.

---

### Meta-Review · Area_Chair_gRVp · 2023-11-22

**Metareview:**

This paper explores collaborative edge caching as a cost-effective solution for improving video caching and streaming. It identifies spatio-temporal diversities in request patterns as the key factor affecting caching performance. To address sparse requests in specific Edge Base Stations (EBSs), the paper proposes knowledge distillation, introducing an adaptive KD-based framework, KDCdCEC. The framework includes a reinforcement learning agent for caching decisions, a deep deterministic policy gradient algorithm for adaptive knowledge distillation, and a content-aware request routing mechanism. Experimental results demonstrate the effectiveness of KDCdCEC in a specific caching setting.

Strengths:
1. A novel approach to collaborative edge caching is explored.
2. This work applies knowledge distillation to facilitate the training of multi-agent RL in addressing the challenges posed by request heterogeneity and dynamics
3. Real dataset is used and some insights from data are provided

Weaknesses:
1. The paper lacks clarity in defining the problem statement for collaborative edge caching, and the assumption of sharing caching policies rather than cached content raises issues.
2. The paper's assumption of a fixed number of video contents deviates significantly from real caching systems, and assuming CDN servers have cached all videos may be overly stringent, considering the substantial number mentioned.
3. The paper lacks details on the training location, process, convergence speed, and time of RL agents and neural networks.
4. The related work section overlooks relevant literature on multi-agent RL and multi-agent RL with knowledge distillation, which could provide valuable context and comparisons.
5. The DRL formulation may not capture key factors of streaming system dynamics, including different video file-size distribution and EBS heterogeneities.
6. The reward design seems to capture only local rewards without a global term, and there is no convergence analysis provided.

**Justification For Why Not Higher Score:**

The paper results may not be reproducible, given the data may not be made public.

**Justification For Why Not Lower Score:**

N/A

---

### Decision · Program_Chairs · 2024-01-16

Reject